# Mindfulness Practice versus Physical Exercise in Enhancing Vitality

**DOI:** 10.3390/ijerph20032537

**Published:** 2023-01-31

**Authors:** Wei Yan, Zhongxin Jiang, Peng Zhang, Guanmin Liu, Kaiping Peng

**Affiliations:** 1Department of Psychology, School of Social Sciences, Tsinghua University, Beijing 100084, China; 2Stanford Graduate School of Education, Stanford University, Stanford, CA 94305, USA; 3Applied Psychology Programme, The Chinese University of Hong Kong, Shenzhen 518712, China; 4Institute of Applied Psychology, Tianjin University, Tianjin 300350, China

**Keywords:** vitality, mindfulness, physical exercise, energy

## Abstract

Vitality is important for subjective well-being and performance, which makes strategies for its enhancement an important research issue. While prior research showed that mindfulness practice and physical exercise are both effective at enhancing vitality, no study has compared their efficacy. This study aimed to address this issue. Seventy-one Chinese adults participated in the study and were randomized to one of the intervention groups, i.e., mindfulness practice or physical exercise. The mindfulness practice group completed guided mindfulness trainings, while the physical exercise group completed self-chosen aerobic trainings for seven days. The levels of vitality and its four factors at three time points (baseline, post-intervention, 7-day follow-up) were measured and compared. Compared with physical exercise, mindfulness practice showed stronger effects in enhancing vitality and maintaining the improvements. The findings suggest that guided mindfulness practice is more effective than self-chosen aerobic physical exercise at enhancing vitality and maintaining its improvements.

## 1. Introduction

Vitality is a positive feeling of having physical and psychological energy available to oneself, which is associated with the feeling of vigor, an activated positive affect, and other positive energized states [1,2]. This construct not only concerns the energy itself or the activation of it but also represents the positive states in which one is able to regulate and direct their available energy to purposive actions [2,3,4]. Therefore, vital individuals appear to be enthusiastic, act positively, and display a spirited outlook with aliveness [1,2]. Unsurprisingly, this subjective experience of vitality is linked with better health and behavioral outcomes. For example, vital people tend to have more positive emotions and a higher satisfaction with life [5,6,7], and more likely to devote themselves to work or study, and achieve better job or academic performance [8,9,10]. These positive outcomes make strategies for enhancing vitality an important research issue.

Physical exercise has long been applied to improve physical and psychological well-being [11,12], and mindfulness has gained popularity in recent decades due to its easy practice and various positive outcomes [13,14]. While both practices are effective at enhancing vitality [11,12,13,14], no studies have hitherto compared their efficacy. This study aimed to address this issue. A comparison of the two popular practices may not only have implications on the psychological mechanisms of vitality but also help individuals or clinicians in choosing suitable practices for enhancing vitality.

Although no studies have compared the two interventions regarding their enhancement of vitality, some recent studies comparing their effects on other health outcomes have begun to shed light on this topic. For instance, it has been found that mindfulness meditation was more effective than physical exercise to help undergraduate nursing students manage their depression [15]. The findings of their comparison in stress reduction were mixed. Some showed that they were equally effective [16], while others found that physical exercise [17] was more effective at reducing stress. On the other hand, a recent study found that yoga was more effective than aerobic exercise at alleviating stress, while mindfulness mediated the effect [18]. Similar mixed findings were found for the effects of these interventions on psychosocial vs. physical-health-related outcomes. For example, Tang et al. [19] compared the effects of long-term mindfulness practice and physical exercise among the elderly and found that the mindfulness practice group showed higher self-ratings for both the physiological and psychological domains of life quality than the physical exercise group. However, Edwards and Loprinzi [20] reviewed five randomized controlled trials comparing meditation and exercise and found that meditation is superior in improving psychosocial variables while exercise in improving variables that were physical health-related. These mixed findings may be due to differences in interventional types, intensity, and/or doses. These comparative studies remain informative for the current research though they were not focused on vitality, because the different effects of mindfulness practice and physical exercise on these outcomes render the speculation that the two interventions may show different effects and mechanisms for enhancing vitality, especially given that these outcomes are associated with vitality [21,22] and that changes in these health outcomes are accompanied by the enhancement of vitality [23].

It is worth noting that most of the studies investigating vitality were conducted in Western societies and measured vitality with the Subjective Vitality Scale (SVS) [1] or its short forms [24], which reflects the Western view of the vitality construct [1,2], i.e., a sole focus on the core component of energy. In Chinese philosophy, vitality is largely reflected in the concept of “Qi”, which is described as an extremely basic and subtle substance with a strong vitality that flows in the human body and is a driving force for inspiring and maintaining human life activities [25]. This leads to a distinctive view of vitality in Chinese people, which not only underscores the feeling of an energy force itself but also the ways in which it is regulated to keep vitality constant. The latter component is the crucial aspect of the Chinese view of vitality, which differs from the Western view of vitality. To conceptualize Chinese vitality, Yan et al. [26] studied the structure of the vitality construct in a Chinese population by interviewing people about their understanding of vitality; by coding the interview transcripts, the authors found that vitality consisted of four factors, i.e., energy (the feeling of the physical and mental energy available to self), tenacity (i.e., volitional and behavioral perseverance), serenity (i.e., balanced utilization of the energy available to oneself), and acuteness (i.e., alertness to inner and outer changes).

Although the four-factor model of vitality is specific to the Chinese view of vitality, its components—energy, tenacity, serenity, and acuteness—are nonetheless closely associated with the Western view of vitality. Yan et al. [26] developed the Four-Factor Vitality Scale (FFVS; the items included on the scale and more detailed information about it can be found in the Appendix A). They found that the total score of the FFVS was strongly correlated with the SVS score (*r* = 0.73). The SVS was particularly highly correlated with the energy factor (*r* = 0.82), which suggests that the energy factor in the FFVS can be considered as being identical to subjective vitality. To put it differently, the feeling of energy is the culturally universal component in both the Chinese and Western views of vitality. On the other hand, Yan et al. [26] also found relatively smaller correlations between the SVS score and three factors—0.62 (tenacity), 0.46 (serenity), and 0.24 (acuteness). That said, though the factors of acuteness, tenacity, and serenity are more culturally specific to the Chinese population, they are closely related to the Western view of vitality. In addition, it was found that energy is closely associated with tenacity or persistence [2,27]. Calmness was proposed as one dimension of vitality [3,4], and one of the items in the SVS measures alertness, i.e., “I nearly always feel alert and awake” [1], which are terms that are closely associated with serenity and acuteness, respectively. Taken together, considering the close associations between the Chinese and Western views of vitality and the increasing acquaintance with and acceptance of “Qi” (as a vital force) in the West—particularly in alternative medical approaches [25,28]—the concept of the four-factor vitality that assimilates the Chinese classical teaching of “Qi” and investigations of it may be of interest and have significant implications for both Chinese and Western populations.

The main objectives of this study were to compare (1) the efficacy of mindfulness practice and physical exercise in enhancing vitality and (2) its four factors (i.e., energy, tenacity, serenity, acuteness) in the Chinese population with the instrument of FFVS. We did not have specific hypotheses about which intervention would show better effects on enhancing vitality and its factors because of the lack of prior investigations comparing the effects of these interventions on vitality.

## 2. Materials and Methods

### 2.1. Participants

A total of 76 Chinese adults participated in the study. They were students from Executive MBA programs (all of them were administrative members in companies), recruited via online advertisements posted on social media. This group of participants was selected, instead of college students, particularly due to their wider age range, which would increase generalizability of the findings to a broader population. They were randomly assigned to either the mindfulness practice (MP) group or the physical exercise (PE) group. One participant from the MP group and four from the PE group dropped out, resulting in a sample size of 71: the MP group included 37 participants (12 males, 25 females) with a mean age of 42.65 ± 6.91 years (age range 25–61 years), and the PE group included 34 participants (12 males and 22 females) with a mean age of 37.56 ± 9.42 years (age range 24–59 years). All participants gave written informed consent (by scanned copy or photographed copy).

### 2.2. Procedure

Before the intervention, all participants completed a baseline vitality measure. Afterwards, all participants were instructed to complete a 7-day intervention. The choice for seven days of continuous training was driven by prior research showing reliable effects with similar [29] or much shorter lengths [18,30]. For the MP group, participants completed a one-hour mindfulness meditation training each day, guided by an experienced mindfulness trainer via videotape delivered online. The daily training contents are shown in Table 1. For the PE group, because some types of physical exercise other than aerobic ones (such as strength training) would decrease one’s vitality [31], participants in this study were instructed to freely choose any kind of aerobic exercises they preferred and to perform it for one hour each day. The decision of guided practice for the MP group but self-chosen activities for the PE group was made because most Chinese people have much less experience in mindfulness meditation (Tai Chi and calligraphy are more traditional and common mental practices among Chinese people [32]; not until recently did mindfulness meditation begin to become popular in China [33]) compared to physical exercise, which is compulsory for all school students. At the end of the last intervention session, all participants completed a post-intervention measure of vitality. A follow-up measure of vitality was also conducted seven days after the last intervention session to investigate the maintenance of any enhancing effects. Participants who completed all tasks obtained two books on psychology (as rewards). The study was conducted in March 2021, and the study protocol was approved by the Institutional Review Board of Tsinghua University.

### 2.3. Measure

In this study, the Four-Factor Vitality Scale (FFVS) [26] was utilized to measure participants’ vitality. This 17-item scale measures four factors of vitality: energy (5 items; e.g., “I feel alive.”), tenacity (4 items; e.g., “I don’t give up easily.”), serenity (4 items; e.g., “I have a mild temper.”), and acuteness (4 items; e.g., “I’m sensitive to changes in surrounding environment.”), using a seven-point Likert scale (1 = strongly disagree; 7 = strongly agree). The higher the total score and the factor scores are, the higher the levels of vitality and its factors are. In its initial development and validation, the FFVS demonstrated good psychometric properties among Chinese adults [26]; by using principal component analysis, the authors confirmed the four-factor structure of the scale, with 69.66% of the total variance being explained (Appendix A). A confirmatory factor analysis also supported the scale structure, showing a good fit of the model (i.e., *χ*^2^ = 284.22, *df* = 113, *χ*^2^/*df* = 2.52 < 3.00, TLI = 0.93 > 0.90, CFI = 0.94 > 0.90, SRMR = 0.05 < 1.00, RMSEA = 0.07 < 0.08; Appendix A). Evidence of criterion validity was provided by the measures of subjective vitality (Appendix A), courage (Appendix A), life satisfaction, positive mental health, resilience, depression, and loneliness (Appendix A). Good internal consistency was shown, with Cronbach’s αs of 0.90 (total score), 0.92 (energy subscale), 0.84 (tenacity subscale), 0.81 (serenity subscale), and 0.78 (acuteness subscale). Good two-week test–retest reliability was also displayed, with coefficients of 0.82 (total score), 0.90 (energy subscale), 0.78 (tenacity subscale), 0.85 (serenity subscale), and 0.76 (acuteness subscale). More detailed information about the scale can be found in the Appendix A. In the current study, the scale showed good internal consistency, with Cronbach’s αs of 0.93 (total score), 0.94 (energy subscale), 0.87 (tenacity subscale), 0.86 (serenity subscale), and 0.87 (acuteness subscale).

### 2.4. Statistical Analysis

Statistical analyses were conducted with SPSS 26.0 (IBM Corp, Armonk, NY, USA) [34]. Independent samples t-tests were first conducted to determine if there were any differences in the baseline levels of vitality and its factors between the two groups. Afterwards, a 2 (group) × 3 (time point) mixed-design repeated-measures ANOVA [35] was conducted to compare the total vitality score at three time points for the two groups (i.e., T1: baseline; T2: post-intervention; T3: 7-day follow-up). To further interpret any group by time point interactions, a series of t-tests were conducted to examine possible group differences between baseline and post-intervention (i.e., T2–T1), between post-intervention and 7-day follow-up (i.e., T3–T2), and between baseline and 7-day follow-up (i.e., T3–T1). The same procedures were repeated to investigate the effects of mindfulness practice and physical exercise on each of the four vitality factors.

## 3. Results

### 3.1. Baseline Characteristics

No significant differences were detected between the two groups for the baseline levels of vitality and its factors (*p*s ≥ 0.25).

### 3.2. Effects of Mindfulness Practice and Physical Exercise on Vitality

There was no main effect of the group: *F*(1, 69) = 1.04, and *p* = 0.31. However, there was a significant main effect of the time point was found on the vitality score: *F*(2, 138) = 82.07, *p* < 0.001, and *η*_p_^2^ = 0.54. Specifically, the vitality score was significantly higher at T2 (*M* = 102.37, *SD* = 11.08) and T3 (*M* = 97.85, *SD* = 11.95) than at T1 (*M* = 92.30, *SD* = 13.95), and the vitality score at T2 was significantly higher than at T3 (i.e., T2 > T3 > T1; *p*s < 0.001). The interaction effect between the group and the time point was significant: *F*(2, 138) = 21.83, *p* < 0.001, and *η*_p_^2^ = 0.24 (Table 2; Figure 1A). Further analyses showed that the increase from T1 to T2 was larger in the MP group (*M* = 12.11, *SD* = 9.13) than in the PE group (*M* = 7.85, *SD* = 5.42): *t*(69) = 2.36, and *p* = 0.019. The decrease from T2 to T3 was smaller in the MP group (*M* = 1.65, *SD* = 4.99) than in the PE group (*M* = 7.65, *SD* = 5.60): *t*(69) = 4.77, and *p* < 0.001. The increase from T1 to T3 was larger in the MP group (*M* = 10.46, *SD* = 8.95) than in the PE group (*M* = 0.21, *SD* = 1.97): *t*(69) = 6.79, and *p* < 0.001 (Figure 1B).

### 3.3. Effects of Mindfulness Practice and Physical Exercise on Factors of Vitality

Energy. The main effect of the group on the energy score was not significant: *F*(1, 69) = 0.73, and *p* = 0.40. However, the main effect of the time point was significant: *F*(2, 138) = 59.71, *p* < 0.001, and *η*_p_^2^ = 0.46. Specifically, the energy score was significantly higher at T2 (*M* = 29.56, *SD* = 4.30) and T3 (*M* = 27.73, *SD* = 4.65) than at T1 (*M* = 26.20, *SD* = 5.57), and the energy score at T2 was significantly higher than at T3 (i.e., T2 > T3 > T1; *p*s < 0.001). A significant interaction effect was found between the group and time point: *F*(2, 138) = 12.46, *p* < 0.001, and *η*_p_^2^ = 0.15 (Table 2; Figure 2A). Further analyses showed that the increase from T1 to T2 was statistically equal between groups (MP: *M* = 3.16, *SD* = 3.20; PE: *M* = 3.59, *SD* = 2.86): *t*(69) = 0.58, and *p* = 0.56. The decrease from T2 to T3 was smaller in the MP group (*M* = 0.46, *SD* = 1.85) than in the PE group (*M* = 3.32, *SD* = 2.47): *t*(69) = 5.56, and *p* < 0.001. The increase from T1 to T3 was larger in the MP group (*M* = 2.70, *SD* = 3.16) than in the PE group (*M* = 0.26, *SD* = 1.38): *t*(69) = 4.15, and *p* < 0.001 (Figure 3A).

Tenacity. The results showed no significant main effect of the group: *F*(1, 69) = 0.01, and *p* = 0.97. However, there was a significant main effect of the time point on the tenacity score: *F*(2, 138) = 28.33, *p* < 0.001, η_p_^2^ = 0.29. There were significantly higher tenacity scores at T2 (*M* = 24.85, *SD* = 2.97) and T3 (*M* = 24.04, *SD* = 3.20) than at T1 (*M* = 23.10, *SD* = 3.85), and the tenacity score at T2 was significantly higher than that at T3 (i.e., T2 > T3 > T1; ps < 0.001). A significant interaction effect between the group and time point was found: *F*(2, 138) = 9.27, *p* = 0.001, and *η*_p_^2^ = 0.12 (Table 2; Figure 2B). Further analyses showed that the increase from T1 to T2 was larger in the MP group (*M* = 2.38, *SD* = 2.78) than in the PE group (*M* = 1.06, *SD* = 1.63): *t*(69) = 2.46, and *p* = 0.017. The decrease from T2 to T3 was statistically equal between groups (MP: *M* = 0.51, *SD* = 1.39; PE: *M* = 1.12, *SD* = 1.45): *t*(69) = 1.79, and *p* = 0.077. The increase from T1 to T3 was larger in the MP group (*M* = 1.86, *SD* = 2.57) than in the PE group (*M* = −0.06, *SD* = 0.81): *t*(69) = 4.32, and *p* < 0.001 (Figure 3B).

Serenity. The main effect of the group on the serenity score was not significant: *F*(1, 69) = 2.13, *p* = 0.15. However, there was a main effect of the time point on the serenity score: *F*(2, 138) = 49.47, *p* < 0.001, and *η*_p_^2^ = 0.42. There were significantly higher serenity scores at T2 (*M* = 23.70, *SD* = 3.49) and T3 (*M* = 22.75, *SD* = 3.74) than at T1 (*M* = 20.97, *SD* = 4.48), and the serenity score at T2 was significantly higher than that at T3 (i.e., T2 > T3 > T1; *p*s < 0.001). Moreover, the interaction effect between the group and time point was significant: *F*(2, 138) = 19.12, *p* < 0.001, and *η*_p_^2^ = 0.22 (Table 2; Figure 2C). Further analyses showed that the increase from T1 to T2 was larger in the MP group (*M* = 3.73, *SD* = 3.24) than in the PE group (*M* = 1.65, *SD* = 2.01): *t*(69) = 3.28, and *p* = 0.002. The decrease from T2 to T3 was smaller in the MP group (*M* = 0.35, *SD* = 1.27) than in the PE group (*M* = 1.62, *SD* = 1.72): *t*(69) = 3.50, and *p* = 0.001. The increase from T1 to T3 was larger in the MP group (*M* = 3.38, *SD* = 3.34) than in the PE group (*M* = 0.03, *SD* = 0.90): *t*(69) = 5.86, and *p* < 0.001 (Figure 3C).

Acuteness. There was no significant main effect of the group on the acuteness score: *F*(1, 69) = 0.70, and *p* = 0.41. However, there was a significant main effect of the time point: *F*(2, 138) = 41.75, *p* < 0.001, and *η*_p_^2^ = 0.38. Specifically, the acuteness score was significantly higher at T2 (*M* = 24.25, *SD* = 2.74) and T3 (*M* = 23.32, *SD* = 3.00) than at T1 (*M* = 22.03, *SD* = 3.87), and the acuteness score at T2 was significantly higher than that at T3 (i.e., T2 > T3 > T1; *p*s < 0.001). The interaction effect between the group and time point was significant: *F*(2, 138) = 13.89, *p* < 0.001, and *η*_p_^2^ = 0.17 (Table 2; Figure 2D). Further analyses showed that the increase from T1 to T2 was larger in the MP group (*M* = 2.84, *SD* = 2.98) than in the PE group (*M* = 1.56, *SD* = 1.78): *t*(69) = 2.22, and *p* = 0.030. The decrease from T2 to T3 was smaller in the MP group (*M* = 0.32, *SD* = 1.33) than in the PE group (*M* = 1.59, *SD* = 1.65): *t*(69) = 3.56, and *p* = 0.001. The increase from T1 to T3 was larger in the MP group (*M* = 2.51, *SD* = 2.61) than in the PE group (*M* = 0.03, *SD* = 0.97): *t*(69) = 5.53, and *p* < 0.001 (Figure 3D).

## 4. Discussion

This study compared the efficacy of mindfulness practice and physical exercise in enhancing vitality and its factors. The results showed that compared to self-chosen aerobic physical exercise, guided mindfulness practice by an expert was more effective at enhancing vitality and the factors of tenacity, serenity, and acuteness as well as in maintaining improvements in vitality and the factors of energy, serenity, and acuteness.

We found that guided mindfulness practice was more effective than self-chosen physical exercise in enhancing vitality and most of its factors. According to the model of vitality based on self-determination theory [2,36], activities that satisfy the basic psychological needs for relatedness, competence, and autonomy enhanced vitality and energy. Both mindfulness practice and physical exercise satisfy basic psychological needs [37,38]. However, it may be possible that mindfulness practice better satisfies basic psychological needs than physical exercise which, thus, is more effective at enhancing vitality. Considering that the interventions compared in this study are guided in one and self-chosen in the other, it is noteworthy that they may have satisfied different aspects of basic psychological needs. While self-chosen physical exercise may have satisfied the need of autonomy more, guided mindfulness may have satisfied the need of relatedness more. Future studies are needed to examine these assumptions.

An alternative explanation for the stronger effect by mindfulness practice is that it might be more effective at strengthening parasympathetic regulation abilities, which are closely associated with vitality [39]. Compared to the elderly who have physical exercise experience, those with long-term mindfulness practice experience showed stronger parasympathetic regulation abilities [20], and practice of the latter—such as heart rate variability biofeedback—effectively enhanced vitality [40]. Nevertheless, it is noteworthy that findings among the elderly may not apply to younger people, such as in our study [41], and a 7-day intervention may not be enough to cause any change in parasympathetic regulation abilities. Another potential physiological mechanism is sleep quality. The relationship between trait mindfulness and subjective vitality is mediated by sleep quality [42], which is associated with parasympathetic cardiovascular modulation [43]. Nevertheless, more direct investigations into the proposed physiological and psychological mechanisms underpinning the improving effects of 7-day mindfulness practice on vitality are required.

Moreover, we found that mindfulness practice was more effective at maintaining the improvements in vitality and most of its factors. Compared to the elderly who regularly exercise, those practicing mindfulness showed stronger neuro-functional connectivity associated with self-regulation [19], and a 5-day mindfulness intervention was enough to increase the ventral midfrontal system’s control over parasympathetic activity [44]. It is possible that the better-maintained effects of mindfulness practice on vitality and its factors were due to these effects being more effective at strengthening the regulation-related neuro-functional connectivity, which may exert effects that have a longer duration. An alternative explanation is that mindfulness practice might have helped those in the mindfulness group cultivate a habit of mindful thinking that persisted even after the intervention was over [45], which, thus, maintained the enhancing effects better, while a behavioral habit of physical exercise was more difficult to establish and, thus, showed a weaker effect on maintaining the improvements. Nevertheless, it is also possible that the difference in this lingering effect in the current findings is due to the fact that physical exercise was self-chosen and mindfulness was guided. The guidance may have provided participants with a structure to be used on their own even after the intervention ended. Future studies should rule out this possibility.

Contrary to the other factors, we found that mindfulness practice and physical exercise were equally effective at enhancing the energy factor of vitality. It is worth noting that mindfulness practice did show a better effect on maintaining the improvements even for the energy factor than physical exercise, which suggests that mindfulness practice is a more recommendable intervention for enhancing vitality and even its core component, the energy factor. Considering that the energy factor is culturally universal to the vitality construct and that our findings showed the enhancing effects of both interventions on the energy factor, like most previous studies [46,47], the conclusion that mindfulness practice is more recommendable than physical exercise for enhancing vitality is likely generalizable to Western populations, though more direct examinations are required.

On the other hand, though mindfulness showed better effects on enhancing the tenacity factor than physical exercise in the present study, both interventions were equally effective at maintaining its improvements. This may be because continual trainings or at least longer intervention programs are necessary for maintaining the enhancing effects on tenacity [48]. It should be noted that the effects of mindfulness practice on maintaining the gains of tenacity were marginally larger than those of physical exercise before multiple corrections, and, thus, the non-results may be not high enough to detect the difference. Future reproductions with a larger sample size are necessary to address the question.

The present study also has practical implications for individuals and clinicians in choosing suitable practices for enhancing vitality. According to the findings, guided mindfulness practice is more recommendable than self-chosen physical exercise in enhancing vitality, especially for those who fail to take exercise because of a lack of time [49], due to its accessibility and easy practice [19]. For those who prefer physical exercise, it may be helpful for them to incorporate components of mindfulness into their exercise protocols or exercise in a way that combine both components, such as yoga [19].

There are some limitations to this study. First, the two interventions were not matched on guidance or intensity. Though previous research has shown that a self-directed mindfulness program is equally effective as an expert-guided one [50], future studies with stricter matching are needed. Second, the participants were not matched on familiarity or habits regarding the corresponding intervention. Since most people have more experience with physical exercise than mindfulness, the present findings may simply result out of a familiarity effect (i.e., participants in the mindfulness group showed a stronger effect simply because they were novices with respect to mindfulness). While familiarity with the two interventions was difficult to match, future studies should at least match participants with a habit of corresponding interventions and include a waitlist control group or even an additional group assigned with an intervention combining mindfulness and physical exercise. Third, as the first study that compared the effects of mindfulness practice and physical exercise on vitality, this study focused on the efficacy of the interventions instead of their mechanisms. Further investigations should study the mechanisms underlying the different effects, especially the roles of the satisfaction of basic psychological needs and parasympathetic regulatory abilities. To achieve the latter goal, physiological indices such as heart rate variability and skin conductance response should be included, which would also warrant more comprehensive comparisons for the two interventions to be distinct from each other.

## 5. Conclusions

Guided mindfulness practice is more effective than self-chosen aerobic physical exercise in enhancing vitality and maintaining its improvements. This finding has implications on the selection of interventions to enhance vitality.

## Figures and Tables

**Figure 1 ijerph-20-02537-f001:**
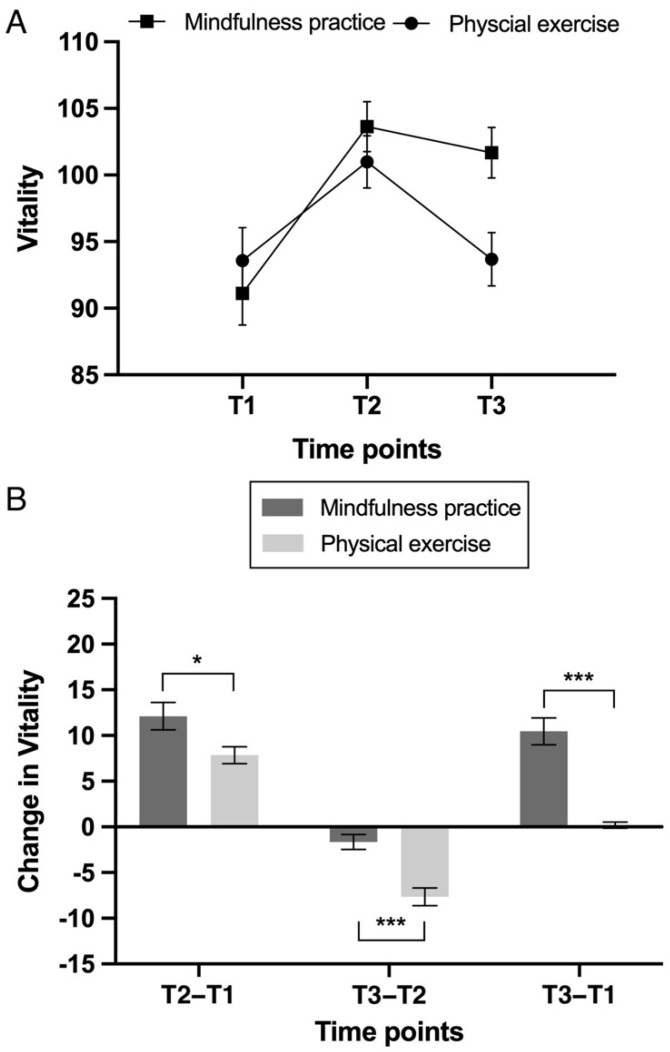
(**A**) Levels of vitality (total score) in the mindfulness practice and the physical exercise groups at three time points. (**B**) Changes in vitality (total score) across different time points (from T1 to T2, from T2 to T3, and from T1 to T3) in the mindfulness practice and the physical exercise groups. *** *p* < 0.001; * *p* < 0.05.

**Figure 2 ijerph-20-02537-f002:**
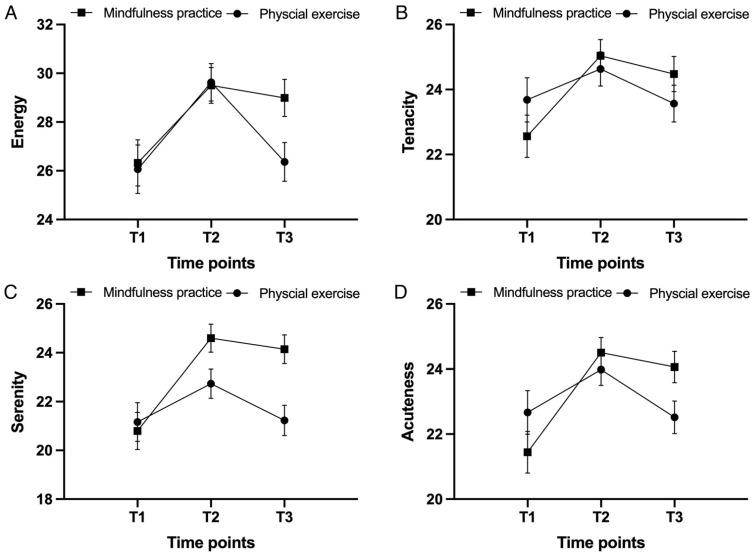
Levels of factors of vitality, i.e., (**A**) energy, (**B**) tenacity, (**C**) serenity, and (**D**) acuteness, for the mindfulness practice group and the physical exercise group at three time points.

**Figure 3 ijerph-20-02537-f003:**
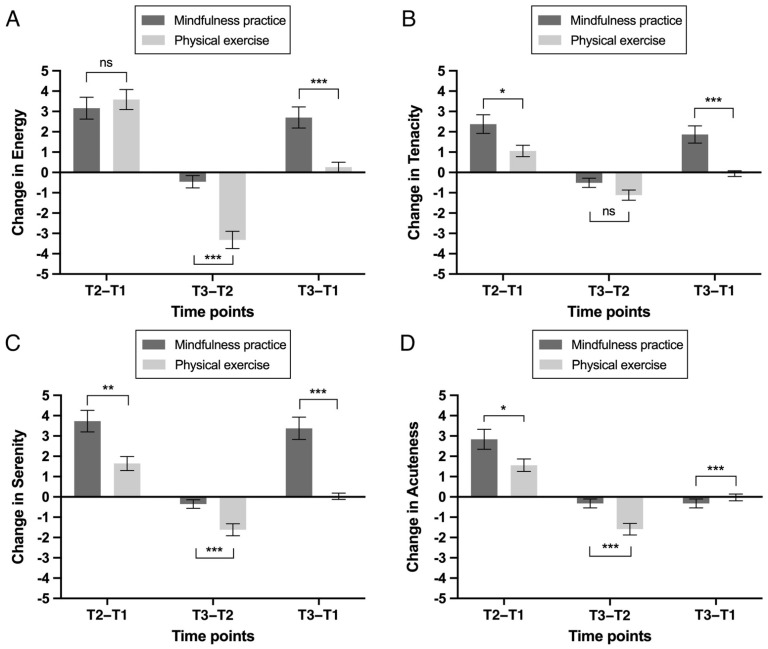
Changes in factors of vitality, i.e., (**A**) energy, (**B**) tenacity, (**C**) serenity, and (**D**) acuteness, across different time points (from T1 to T2, from T2 to T3, and from T1 to T3) in the mindfulness practice group and the physical exercise group. *** *p* < 0.001; ** *p* < 0.01; * *p* < 0.05; ns, not significant.

**Table 1 ijerph-20-02537-t001:** The contents of daily mindfulness meditation training.

Day	Guided Practice
1	Body-scan meditation and breath watching
2	Mindful focus on present thoughts and feelings
3	Sitting meditation and walking meditation
4	Mindful acceptance of thoughts without judgment
5	Loving-kindness meditation
6	Mindful acceptance of experiences in daily life
7	Review of practices learned in the previous six days

**Table 2 ijerph-20-02537-t002:** Levels of vitality and its factors and the interaction effects between group and time point.

	Mindfulness Practice Group (*n* = 37)			Physical Exercise Group (*n* = 34)			Interaction Effect	
	T1	T2	T3	T1	T2	T3	*F*(2, 138)	*η* _p_ ^2^
Vitality (total score)	91.32	103.43	101.78	93.35	101.21	93.56	21.83 ***	0.24
	(16.14)	(11.36)	(12.03)	(11.25)	(10.81)	(10.42)		
Energy factor	26.32	29.49	29.03	26.06	29.65	26.32	12.46 ***	0.15
	(6.10)	(4.80)	(4.78)	(5.01)	(3.76)	(4.13)		
Tenacity factor	22.59	24.97	24.46	23.65	24.71	23.59	9.27 ***	0.12
	(4.69)	(3.18)	(3.56)	(2.62)	(2.76)	(2.74)		
Serenity factor	20.70	24.43	24.08	21.26	22.91	21.29	19.12 ***	0.22
	(4.60)	(2.88)	(3.04)	(4.39)	(3.95)	(3.94)		
Acuteness factor	21.70	24.54	24.22	22.38	23.94	22.35	13.89 ***	0.17
	(4.62)	(2.78)	(2.96)	(2.87)	(2.71)	(2.77)		

*** *p* < 0.001. The numbers in the parentheses are standard deviation of scores for each group and time point.

## Data Availability

The data presented in this study are available on request from the corresponding authors.

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
