# Peer review of "Mindfulness Practice versus Physical Exercise in Enhancing Vitality"

_ijerph, 2023, doi:10.3390/ijerph20032537_

Round 1

Reviewer 1 Report (Previous Reviewer 2)

The authors are thanked for their resubmission and for taking into consideration the suggested edits of this reviewer in making the changes for the resubmission. The strengths of this paper remain that it is well-written and original. Furthermore, it is well-conceptualized. Each of the changes that have been made have improved the paper. 

In reading over the submission again, this reviewer has noted that why and how vitality is discussed as it is in the paper should be improved. The authors begin with the 1997 understanding of vitality by Ryan et al. and then backtrack that how vitality is understood in China differs from this understanding. As such, the authors have to explain why they started in this way, rather than instead with the understanding of vitality as Qi. The specific problems with how the Introduction is structured are outlined in the line by line edits below.

One thing this reviewer commented on in the previous review (with respect to line 46) that was not expanded on sufficiently by the authors is the information about the COVID-19 lockdown and how this relates to vitality. Either the authors are concerned about the reduction in vitality as a result of the COVID-19 lockdowns or they are not. If they are concerned, then this should be added to the title and COVID-19 should become one of the keywords. Furthermore, there needs to be information provided regarding research on the relationship between COVID-19 and vitality.

The authors are also advised to incorporate all of the information in the Supplementary Materials into the body of the paper. In the line by line edits below, suggestions are provided regarding how the material can be separated to have some of it appear in the Introduction and the rest in the Procedure and Materials.

The authors are also reminded that unless research is seminal in the area, only research conducted within the last five years should be cited. With respect to the Introduction, references 1 and 3 can be considered seminal and only references 8,11,12,13,16,17,19,26-31 are current. Please find more current references to replace the others that are out of date. Similarly, for the Discussion, the only current references are 37 and 47. All the other references are out of date and need others to replace them that report on current research. When authors are conducting cutting-edge research, as these authors are, the theoretical foundations of the Introduction and Discussion must be current.

The authors should also note that, according to MDPI style for references, all references require a DOI number. Please include the DOI number for each reference.

Other than these major points, there are some minor issues that are mentioned below, in the line by line edits.

Line by line edits.

2 Add “, Especially During the COVID-19 Pandemic Lockdown” or a similar phrase regarding the COVID-19 Pandemic Lockdown.

23 Add “COVID-19”.

28 Change “state” to “states”.

56-57 In this section where the authors have been comparing physical exercise to meditation regarding stress what does it mean that the “mindfulness group showed higher self-ratings for the physical domain of life quality”? Are the authors meaning that the study only concerned the physical domain and did not mention stress?  In reading over the Tang et al. (2020) reference, stress is mentioned. Here is a quotation from that article, “Studies have shown that 5 days of mindfulness meditation using integrative body-mind training (IBMT) improves attention and reduces stress reactivity through changing the interaction between the anterior cingulate cortex (ACC) and the parasympathetic branch of the autonomic nervous system (Tang et al., 2007, 2009).” Further on, Tang et al. (2020) includes this statement, “It should be noted that our previous research has reported mindfulness effects on attention, cognitive performance, emotional states and stress regulation. In the current study, we aimed to further explore physical and psychological changes of quality of life using widely validated WHO Quality of Life Survey. Future research should include multi-faceted questionnaires to fully evaluate the potential changes.”. As such, the authors should point out that Tang et al. (2020) have found that mindfulness has a positive effect on stress but their most recent research has been focused on the physical effects of meditation. Yet, Tang et al. (2020) is supportive of future research regarding stress. The research undertaken by the authors is of this type. The authors need to indicate why it is important to reference the 2020 study by Tang et al. in this section of their paper concerned with stress. As such, these quotations from Tang et al. (2020) provide a reason, and they should be noted and summarized by the authors.

58-59 Change “they were more effective in psychosocial and physical health-related variables, respectively” to “meditation was more effective in psychosocial variables and exercise in those variables that were physical health-related”.

62 In this paragraph that has primarily concentrated on alleviating stress, there needs to be a definition of stress and some information on how alleviating stress relates to “enhancing vitality”.

67-69 Given that the authors are noting the difference between he Western notion of vitality and that of Chinese culture, it is pertinent to discuss the idea of Qi in Chinese culture rather than merely saying that there is a difference in the meaning of vitality. Western cultures are aware of the term Qi. It would be easier to understand the difference between the two meanings of vitality if the term Qi were defined and used by the authors.

68-75 The article written by the authors cited as 31 is supposedly in press to be published in the Chinese Social Psychological Review. According to a Google search, the last time this journal published was volume 20 in 2021 (http://www.ssapchina.com/SKYW/SelectInfo.aspx?con=Chinese+Social+Psychological+Review). There have been no further publications of this journal reported by the publishers. As a result, rather than citing this research to be in press for volume 23 of this journal, it if the information in 31 were included as part of this current manuscript instead of as supplementary material this would be preferred.

80-81 “the factors of acuteness, tenacity and tenacity are more culturally specific to the Chinese population”. This statement by the authors is only true if the focus is on the word “vitality”. If, instead, the Chinese word, Qi, were used, Western cultures are aware of—and also can focus on—Qi as consisting of energy, tenacity, serenity and acuteness as the idea of Qi is becoming increasingly accepted into Western medicine ( See, for example, https://doi.org/10.3390/healthcare9030257 and https://doi.org/10.1186/s12906-020-03174-1 ). In this phrase I have quoted above from the author’s paper, please note tenacity appears twice and neither energy nor serenity (as the two other factors of vitality) are mentioned. Pleases correct this.

86 Change “Though the factors of” to “Although of the four factors associated with vitality—energy, tenacity, serenity and acuteness—”.

25-101 This Introduction begins with a Western definition of vitality. Given that this is a study conducted in China and the Chinese definition of vitality is assumed by those part of the intervention, this Introduction needs to be rewritten to specify that the understanding of vitality assumed by the participants as Qi. Then Qi needs to be compared with the traditional Western understanding of vitality. However, it also needs to be mentioned that the West is becoming increasingly familiar with Qi—particularly in medical education—and that, as such, it is important to investigate Qi for international audiences, especially with respect to COVID-19 lockdowns. Furthermore, this introduction also has to include research regarding the effect of COVID-19 lockdowns on vitality. Here is a study that may be helpful in this regard: https://doi.org/10.3390/jcm10163516. Part of this rewriting could be to take the first paragraph of “The Theoretical Construction and Measurement of Vitality in the Chinese Cultural Context” from the Supplementary materials and summarize it in the text as part of the Introduction.

105 Why was this group of participants selected? Please add this information. As well, indicate how informed consent was obtained and what was the stimulus for the subjects to participate?

115-121 This is where the detailed information from the Supplementary Material could be inserted rather than referring to it in a supplement.

130-131 It is interesting that most Chinese are so unfamiliar with mindfulness practice that they have to be taught the practice by an expert. In the West, it is thought that, as a country rooted in Buddhism, mindfulness practice would be well-known in China and taught by parents to their children. It would be useful for the authors to mention why mindfulness is not well-known in China.

142 Please provide a reference for SPSS 26.0

145 Please provide a reference for ANOVA.

318 Change “novice” to “novices with respect”

319 Change “were difficult to be matched” to “was difficult to match”.

331-333 The Conclusion needs to mention the importance of this research with respect to COVID-19 lockdowns.

335-340 Again, the suggestion is that the Supplementary Material be incorporated into the body of the paper with the relevant parts added to either the Introduction or the Procedure and Materials.

Author Response

Dear Reviewer,

Many thanks to you for your careful review of our manuscript and your sage advices for improving the quality of it. My co-authors and I really appreciate the care, time, and effort that went into the review process. We are most grateful for the precious opportunity you gave us to revise the manuscript. We have finished the revision, and now we submit it for your review.

Please check the attachment, which is responding letter to reviewer. I listed all the corrections, modification and additions to each of the reviewer comments. We are hoping that you find our responses satisfactory.

We would be grateful if you could kindly advise us if we missed anything important or any additional changes are needed, we would be happy to make additional revision needed to merit your standard for publication in your journal.

Yours sincerely,

Authors.

Reviewer 2 Report (New Reviewer)

Good morning,

I found your paper very well thought out and clearly written. I commend you on your work.

I have several questions which may or may not effect the results: 

* were there any common themes or self-selected exercises chosen by the PE group, or were they very diverse?

* In the PE group, are you aware if any members were already involved in some sort of daily PE prior to participation in the study?

Overall, very well done.

Best of luck!

Author Response

Dear Reviewer,

Many thanks to you for your careful review of our manuscript and your sage advices for improving the quality of it. My co-authors and I really appreciate the care, time, and effort that went into the review process. We have finished the revision, and now we submit it for your review.

Please check the attachment, which is responding letter to reviewer. We are hoping that you find our responses satisfactory.

Yours sincerely,

Authors.

Round 2

Reviewer 1 Report (Previous Reviewer 2)

This is a much improved version. To the extent that the authors did not take up the suggested edits of the reviewer with respect to COVID-19, the reviewer is in complete agreement that mention of COVID-19 is best removed from the paper.  The reviewer also accepts the logic of the authors regarding the older references that were not updated, as they are considered either seminal or of particular importance.

In this reviewer’s view, the paper is now ready for publication after correction of these few items with respect to wording or punctuation.

Line by line suggested edits.

62 Change “that, the” to “that the”.

84 Change “they are” to “are”.

107 Change “in Chinese” to “in the Chinese”.

124 Change “photographical” to “photographed”.

272 Change “is guided” to “are guided”.

281-282 Change “latter, such as heart rate variability biofeedback effectively” to “latter—such as heart rate variability biofeedback—effectively”.

294 Change “mindfulness” to “mindfulness intervention”.

315 Change “factor like” to “factor, like”.

316 Change “might probably be” to “is likely”.

324 Change “may be out of power not high” to “may be not high”.

339 Chane “may simply out” to “may simply result out”.

341 Change “familiarity to” to “familiarity with”.

392 Add “https://doi.org/10.18844/cjes.v15i5.5121”.

407 Add “https://doi.org/10.3389/fpsyg.2018.02090”.

423 Add “https://doi.org/10.3389/fpsyg.2020.00358”.

435 Add “https://doi.org/10.3389/fpsyg.2021.659667”.

453 Add “https://doi.org/10.1159/000095387”.

497 The author needs to add: “Available at https://www.jstor.org/stable/44954053 (accessed on 23 January 2023).

Supplementary

The heading for Table 1S is: “The discrimination analysis for the 21 items”. This heading does not mention what is involved in the discrimination analysis for the 21 items as represented by the five additional columns after the items. Please improve this heading to include this information.

The heading for Table 2S is “The matrix of factor loadings for the 21 items”. This heading does not mention what the different factor loadings represent for the 21 items as represented by the four additional columns after the items. Please improve this heading to include this information.

The heading for Table 3S is “The matrix of factor loadings for the final 17 items”. This heading does not include mention of Energy, Tenacity, Serenity and Acuteness. Please improve this heading to include this information.

Author Response

Dear Reviewer,

Many thanks to you for your very much careful review of our manuscript and your sage advice for improving the quality of it. My co-authors and I really appreciate the care, time, and effort that went into the review process. We have finished revision, and now we submit it for your review.

Yours sincerely,

Authors

This manuscript is a resubmission of an earlier submission. The following is a list of the peer review reports and author responses from that submission.

Round 1

Reviewer 1 Report

The purpose of this study is to examine the effects of mindfulness training on vitality, strength, and serenity compared to physical activity. The topic is interesting, but there are fatal errors in the study design.

1. The author described that mindfulness practice was observed to increase vitality, strength, and comfort compared to physical activity. However, it is judged that the level of physical activity used in this study is time, frequency, intensity, and duration that are not reasonable to obtain the effects of known physical activity.

2. The intensity of the exercise is not properly stated. Also, exercise has different effects depending on intensity, leadership, and encouragement. The time for exercise and mindfulness practice are the same, but for those who have not exercised on a regular basis, 1 hour of exercise can be difficult. It can have enough adverse effects.

3. Researchers conducted aerobic exercise freely. Since it is not possible to check the subjective intensity or indicators of exercise intensity such as heart rate and energy consumption during 1 hour of exercise, comparison between groups cannot be performed.

4. In order to compare the effects of mindfulness training and exercise, it is said that it is a better design to identify a group that does nothing, a group that does mindfulness, a group that does exercise, and a group that combines mindfulness and exercise.

5. As there are no physiological indicators other than subjectively felt indicators, it may confuse readers.

Reviewer 2 Report

This is an original study conducted on 71 Chinese adults to determine which of guided mindfulness or self-chosen exercise had a greater effect of improving vitality after a seven day intervention. The participants were divided into two groups: one to receive guided mindfulness and the other to engage in self-selected exercise. The two groups were tested at three times: baseline, after the seven day intervention, and one week after the intervention. The authors found that guided mindfulness was more effective than physical exercise in enhancing vitality after the intervention and in maintaining improvements after a week.

The strengths of this paper are that it is well written and original. Although there have been studies on both mindfulness and on physical exercise, until this paper, there has been no research comparing which of the two might give better results.  In the supplementary material, the authors have also created their own instrument specific to Chinese populations regarding vitality. If this research has been published somewhere else, this is not indicated. This is important research. My suggestion is that the supplementary material should be published separately, not as merely a supplement to this paper. 

Nevertheless, there were limitations to this study, unmentioned by the authors. Whether or not it makes a difference that the mindfulness activities were guided and the physical education activities were self-chosen was not noted by these authors as a variable to take into consideration (here is a link to a study that looked into the difference between guided and self-selected mindfulness. http://dx.doi.org/10.1136/thoraxjnl-2017-211264. The results would be helpful to the authors in giving support to their not considering this variable). Furthermore, the research the authors refer to most is respect to the elderly; yet, the group they investigated involved younger people. The authors don’t mention that there could be a meaningful difference between the effect of guided mindfulness and of physical exercise on younger people compared with the elderly.  An additional limitation is that the authors don’t consider that the lingering effect of the guided mindfulness may not be because of mindfulness but because of the particular guided work that was done. In effect the guide provided the participants with a structure that they could use on their own and the participants might still be engaged in the mindfulness practice after the end of the study. The authors assumed that after a week the participants would neither be engaged in mindfulness nor in physical exercise. These assumptions might be incorrect, especially regarding mindfulness for the reason mentioned.

The authors don’t say when the study was conducted. This needs to be added. As well, it is stated that those interested in participating reached out to researchers—how did they reach out? The authors don’t say. How did the participants give informed consent? From where did they get Ethic’s approval? Was this research reviewed by a research ethics board? If so, where is the Institutional Review Board Statement? There is no Data Availability Statement.  Please add this at the end of the paper.

As per the MDPI Instructions for Authors, please redo all the citations and references so that they conform to the expected standard. 

Line by line suggested edits

46 “especially during the COVID-19 pandemic lockdown”, please provide further information on why this is especially so during the COVID-19 pandemic lockdown and provide a reference. 

97-99 Change “The main objectives of this study were to (1) compare the efficacy of mindfulness practice and physical exercise in enhancing vitality, and (2) its four factors (i.e., acuteness, tenacity, and serenity) in Chinese population.” to “The main objectives of this study were to compare (1) the efficacy of mindfulness practice and physical exercise in enhancing vitality, and (2) its four factors (i.e., energy, tenacity, serenity and acuteness) in the Chinese population.” 

113-115 Given that the participants were recruited from the China Europe International Business School, it is surprising that the mean age for each of the two groups was so old. Was this unexpected by the authors?  Please include additional information on the type of people attending this school.

124 Why was a seven day intervention chosen? What evidence is there that seven days are sufficient to see a change in vitality from either intervention? Please provide references to this effect.

126 “an experienced mindfulness trainer”, please provide references to support this decision to use an experienced trainer for mindfulness but self-chosen activities for the physical exercise. 

Table 1 

As the heading for the left column is “Day”, delete “Day” from each of the entries so that only the numbers 1-7 appear in the column.

Table 2 

Please redo the spacing of the tables so that the spaces between T1, T2 and T3 for both the Mindfulness Practice Group and the Physical Exercise Group are equal.

 232-233 Change “The results showed that compared to physical exercise, mindfulness practice was more effective in enhancing vitality” to “The results showed that compared to self-chosen physical exercise, guided mindfulness practice by an expert was more effective in enhancing vitality”.

235-237 Change “These findings suggest that mindfulness practice is more effective than physical exercise in enhancing vitality and maintaining its improvements.” to “These findings suggest that guided mindfulness practice by an expert is more effective than self-chosen physical exercise in enhancing vitality and maintaining its improvements.”

238-239 Change “We found that mindfulness practice was more effective than physical exercise in enhancing vitality and most of its factors.” to “We found that guided mindfulness practice was more effective than self-chosen physical exercise in enhancing vitality and most of its factors. 

240 Given that the authors have highlighted self-determination theory, it needs to be stated by the authors that the mindfulness activities, because they were guided, were not self-determined. This being the case, the particular mindfulness activity engaged in does not relate to the research regarding self-determination theory.

249 “those with long-term mindfulness practice”, this information may be of interest, but the authors need to remind readers that their intervention was only for 1 week, so they cannot claim such changes occurred with their participants. 

257 “Compared to elderlies” the authors have mentioned research on elderlies a number of times to this point. Yet, their own study was on younger participants. Whether the results of mindfulness practices with younger people are comparable to an older population is not considered by the authors. Here is a paper that does consider this: DOI: 10.1016/j.concog.2009.05.001 and should be taken into consideration by the authors.

261 Change “It may be possible that the better maintaining effects of mindfulness practice on vitality and its factors were due to that they were more effective” to “It may be possible that the better maintained effects of mindfulness practice on vitality and its factors were due to their being more effective”.

296 “such as yoga and tai chi”, since this is only the second time yoga and the first time tai chi are mentioned in this paper, and neither is the same as mindfulness, references are needed to show that yoga and tai chi are helpful in similar ways to mindfulness training. The Tong et al reference can by cited here for yoga, but a new reference is needed for tai chi.

Supplementary materials

Table S1

Please increase the space between the Vitality and Energy column as Vitality is cumulative of Energy, Tenacity, Serenity and Acuteness. It would be better if the Vitality column followed the Acuteness column, rather than coming before Energy.

FFVS

My assumption is that the authors conducted the study in Chinese and these are English translations (it needs to be stated in what language the study was conducted). As such, I have the following comments.

11. I treat people friendly.” This is not proper English. Depending on what point the authors want to emphasize, it can be worded in one of these ways “11. I am friendly to other people”, “11. I treat people respectfully”, or, “11. I treat other people as if they are my friends”.

“12. I’m emotionally stable mostly.” is very unclear. Better wording would be any of the following, “I am calm most often”, “I consider myself emotionally stable”, “I am considered by others to be emotionally stable”.

“14. I am fond of observing people” is a phrase in English that might be interpreted to mean “I am a voyeur”, which would not be what the authors intend. Better wording in English is “14. I am interested in people”, or “14. I like to take note of what people do”.